# Thermal Transport and Rheological Properties of Hybrid Nanofluids Based on Vegetable Lubricants

**DOI:** 10.3390/nano13202739

**Published:** 2023-10-10

**Authors:** Hélio Ribeiro, Jose Jaime Taha-Tijerina, Ofelia Gomez, Ever Acosta, Gabriel M. Pinto, Lorena R. C. Moraes, Guilhermino J. M. Fechine, Ricardo J. E. Andrade, Jefferson Reinoza, Victoria Padilla, Karen Lozano

**Affiliations:** 1Department of Informatics and Engineering Systems, University of Texas Rio Grande Valley—UTRGV, Brownsville, TX 78520, USA; helio.ribeiro1@mackenzie.br; 2Engineering School, Mackenzie Presbyterian University, Rua da Consolação 896, São Paulo 01302-907, SP, Brazil; gabriel.matheus.pinto@gmail.com (G.M.P.); guilherminojose.fechine@mackenzie.br (G.J.M.F.); ricardo.andrade@mackenzie.br (R.J.E.A.); 3Department of Mechanical Engineering, University of Texas Rio Grande Valley—UTRGV, Edinburg, TX 78539, USA; ofelia.gomezchacon01@utrgv.edu (O.G.); ever.acosta@utrgv.edu (E.A.); jefferson.reinoza@utrgv.edu (J.R.); victoria.padilla@utrgv.edu (V.P.); karen.lozano@utrgv.edu (K.L.); 4Mackenzie Institute for Research in Graphene and Nanotechnologies—MackGraphe, Mackenzie Presbyterian University, São Paulo 01302-907, SP, Brazil; 5Departament of Mechanical Engineering, Pontifícia Universidade Católica do Rio de Janeiro, Católica do Rio de Janeiro 22453-900, RJ, Brazil; lorenarcmoraes@gmail.com

**Keywords:** hybrid nanofluids, thermal conductivity, rheological properties

## Abstract

Nanofluids based on vegetal oil with different wt.% of carbon nanotubes (CNT), hexagonal boron nitride (h-BN), and its hybrid (h-BN@CNT) were produced to investigate the effects of these nano-additives on the thermal conductivity and rheological properties of nanofluids. Stable suspensions of these oil/nanostructures were produced without the use of stabilizing agents. The dispersed nanostructures were investigated by SEM, EDS, XRD, and XPS, while the thermal conductivity and rheological characteristics were studied by a transient hot-wire method and steady-state flow tests, respectively. Increases in thermal conductivity of up to 39% were observed for fluids produced with 0.5 wt.% of the hybrid nanomaterials. As for the rheological properties, it was verified that both the base fluid and the h-BN suspensions exhibited Newtonian behavior, while the presence of CNT modified this tendency. This change in behavior is attributed to the hydrophobic character of both CNT and the base oil, while h-BN nanostructures have lip-lip “bonds”, giving it a partial ionic character. However, the combination of these nanostructures was fundamental for the synergistic effect on the increase of thermal conductivity with respect to their counterparts.

## 1. Introduction

There are several challenges regarding the use of oils derived from fossil fuels due to their depletion and the fact that this resource is limited and non-renewable [1]. In this sense, the extraction of oil and natural gas has become more difficult and expensive with time, which may lead to increased prices and decreased availability of these resources in the future [2]. Moreover, burning fossil fuels emits greenhouse gases such as carbon dioxide (CO_2_), which contribute to global warming and climate change. Furthermore, the extraction and transportation of these fuels can lead to leaks, soil and water pollution, and other detrimental environmental impacts. In addition, the dependence on fossil fuels can hamper the development and adoption of cleaner, renewable alternatives such as solar, wind and hydropower, among others [2].

In this context, the use of vegetable oils, as lubricants and dielectric coolers, has been increasingly explored as an alternative to petroleum-derived oils [3,4]. Vegetable oils are renewable, biodegradable and have a lower environmental impact. Furthermore, some vegetable oils, such as canola, soybean, and sunflower, have lubricating and cooling properties similar, or even superior, to mineral oils [1]. It is important to highlight that the use of vegetable oils in industrial applications is still relatively new and, therefore, there are challenges and limitations to be considered [5]. For example, vegetable oils may have lower thermal and oxidative stability at high temperatures and may be more susceptible to degradation by microorganisms. However, the use of inorganic nanoparticles can minimize these undesirable effects [3]. Despite these limitations, the use of vegetable oils, as lubricants and dielectric coolers, has been explored in applications such as machining, injection molding, metal cutting and cooling of electrical transformers, among others [6].

With technological advances and the search for more sustainable solutions, it is possible that the use of vegetable oils in industrial applications will become increasingly common and viable for applications aimed at preserving the environment. Recent works regarding thermally conductive nanoparticles have shown that it is possible to improve the physical and chemical properties of oils by adding them in low concentrations [7]. In this sense, in order to improve the thermal and rheological performance of oils, nanomaterials with different dimensions can be used, such as carbon nanotubes (CNT) and boron nitride (h-BN) [2,6]. In the case of multi-walled carbon nanotubes, the particles are cylindrical structures composed of multiple concentric layers of graphene. They have unique properties, such as high mechanical strength, electrical and thermal conductivity, in addition to being lightweight and resistant to corrosion [8]. This type of nanomaterial has been used in different composite systems where thermal, rheological and tribological improvements were observed [9]. Likewise, hexagonal boron nitride (h-BN) has been used for the same purposes due to its exceptional physical and chemical properties. Hexagonal boron nitride (h-BN) nanostructures have excellent tribological properties and, due to their high thermal and chemical stability, they are used as a lubricating agent at high temperatures and pressures conditions [3,10,11,12]. This nanomaterial is also an excellent lubricating agent due to its low friction coefficient. h-BN is ecologically friendly and inert to most chemicals, which makes this nanomaterial attractive for many industrial applications [6]. However, one of the biggest challenges encountered in its use is the production of stable dispersions, since natural or synthetic oils are polyolefin based. Besides this, due to the partial ionic bonds (lip-lip) that h-BN exhibits between its layers, which provide greater interlayer attraction, the exfoliation of h-BN from the bulk is substantially more challenging than that of conventional graphene [13]. Thus, due to the lack of functional groups, generally the use of dispersing agents (surfactants) or special pre-dispersion techniques are usually necessary to produce stable emulsions [14].

Some works have already been devoted to the study of the use of hybrid nanoparticles in lubricating oils. For instance, Ahmad et al. [15] investigated the thermal and magnetic characteristics of a nanofluid in motor oil (EO) with graphene oxide (GO), and the hybrid nanofluid GO–Fe_3_O_4_/EO under the same effect of the induced applied magnetic field. They observed that the hybrid GO–Fe_3_O_4_/EO composition provided the thermal stability of the nanofluids at high temperature. Wanatasanappan et al. [16] investigated the thermal conductivity of different coconut, soy, and palm-based vegetable oils suspended with Al_2_O_3_-TiO_2_ hybrid nanostructures at different wt.%. They verified that the hybrid nanofluids’ thermal conductivities increased with temperature and concentration for all three base oils. However, the palm oil-based hybrid nanofluid with 0.6 wt.% of Al_2_O_3_-TiO_2_ showed an increase of thermal conductivity of up to 125%, demonstrating compatibility between Al_2_O_3_-TiO_2_ nanoparticles with the base fluid. It has been reported that the thermophysical properties of nanofluids, such as thermal conductivity and viscosity, strongly depend on temperature, nanoparticle size, concentration, and in some cases on the pH value [17,18]. Nonetheless, the science regarding hybrid fluids is still quite complex in terms of their response to temperature, the nature of the nanoparticles, base fluid, presence/absence of surfactant, dispersion processes, among others.

In this work, nanofluids based on CNT, h-BN and its hybrids’ (h-BN@CNT) nanostructures were produced at different concentrations using a pre-dispersion method in vegetable oil, aiming to study their thermal conductivity and rheological properties. To the best of our knowledge, the use of hybrid nanoparticles with BN@CNT in natural vegetable oils has not yet been reported in the literature.

## 2. Materials and Methods

For this work, a synthetic ester vegetable oil-based lubricant—Envirotemp^®^ FR3™ was used (Cargill Industrial Specialties, Minneapolis). This lubricant does not contain petroleum, halogens, silicones, or corrosive sulfur. The physical and chemical characteristics of the chosen lubricant are shown in Table 1. To create various nanofluids, h-BN, CNT, and h-BN@CNT nanostructures were used as the building blocks in concentrations of 0.10, 0.25, and 0.50 wt.%. The nanofluids with the hybrid compositions were used at a mass ratio of 1:1 between h-BN and CNT. The CNT used in this work are multi-walled CNT from Cheap Tubes (Grafton, VT, USA), and the h-BN was purchased from Sigma-Aldrich (CAS No. 10043-11-5; St. Louis, MO, USA). Table 2 presents the specifications of these nanomaterials. Initially, the CNT were debundled in a mixture of 1:3 dimethylformamide (DMF)/tetrahydrofuran (THF) using bath sonication (Branson ultrasonic homogenizer model 5510, Danbury, CT, USA, 40 kHz) for 5 h. Then, the suspensions were filtered, and the material was dried in an oven for 24 h at 90 °C. The h-BN nanostructures were exfoliated using bath sonication in isopropanol (IPA) for an extensive time (>8 h), filtered, and then dried in an oven for 24 h, according to previous research [3]. After that, the powdered nanomaterials were added in the alcoholic suspension and mixed with the vegetable oil at 80 °C for 12 h until the total evaporation of the alcohol. The samples were kept in a drawer for at least 2 weeks at room temperature without significant sedimentation. Stable homogeneous nanofluids were obtained and subsequently characterized. A Bruker D8 Advance X-ray diffractometer (Bruker Advanced X-ray Solutions [AXS], Edinburg, TX, USA) was used to perform X-ray diffraction (XRD) on the nanostructures applying Cu Kα radiation (λ = 1.5418). The XRD patterns were obtained in the 2θ range between 4° and 80° at a scanning rate of 4° min^−1^. Without using any surfactant agents, the nanostructures were dispersed in isopropanol (IPA) via ultrasonication to produce the h-BN, CNT, and h-BN@CNT samples. After this process, the dispersions were dripped onto a double-sided carbon tape and dried for 12 h in an oven. The morphology of h-BN, CNT and the hybrid nanostructures was investigated by SEM through Field-Emission Scanning Electron Microscopy (FE-SEM, Vp Sigma from Carl Zeiss, Jena, Germany), and SmartSEM v6.01 software (Edinburg, TX, USA). Monochromatic Al Kα radiation (1486.6 eV) and an electron energy analyzer (Specs, Phoibos-150) from Edinburgh, TX, USA, were used to produce X-ray photoelectron spectroscopy (XPS). The vacuum chamber’s base pressure was 2.0 × 10^10^ mbar. The precise oxidation states of each element were determined by fitting the XPS peaks under the assumption that they followed the shape of a convolution of various Lorentzian and Gaussian contributions.

Thermal performance was measured using a TEMPOS thermal analyzer device (METER GROUP, Inc., Pullman, WA, USA) under the transient hot-wire technique. Thermal conductivity was evaluated at room temperature, 50 °C, 75 °C and 90 °C. Each sample was first bath sonicated for 15 min prior to each experimental test, and each set of specimens was kept in thermal equilibrium for at least 15 min prior to measurements of experiments conducted above room temperature. The measured thermal conductivity values were compared to the regular vegetable lubricant (k_0_). The effective thermal conductivity of the nanofluids (k_eff_) was calculated by Equation (1). A minimum of ten measurements were taken for each composition, and the mean values were reported with the standard deviation as error bars.
Enhancement in TC (%) = [(k_eff_/k_0_) − 1](1)

The nanofluids’ rheological behavior was analyzed through steady-state flow tests performed on a Haake Mars 40 (Thermo Scientific) stress controlled rotational rheometer (Waltham, MA, USA) coupled with a P35/Ti parallel plate geometry (35 mm in diameter) and a 0.5 mm gap. Flow and viscosity curves are presented on a logarithmic scale over a shear rate range of 10 to 100 s^−1^. All tests were conducted at room temperature and pressure. Triplicate tests were performed for all samples, to ensure repeatability.

## 3. Results

### 3.1. SEM Images

Figure 1a–c shows SEM images of h-BN (a), CNT (b) and the h-BN@CNT (c) nanostructures at different magnifications. Figure 1c demonstrates that both nanomaterials were mixed to form the hybrid nanostructures. A few small agglomerates are present, but it is also clearly observed how the h-BN particles mixed the CNT bundles apart.

### 3.2. XRD and XPS Analysis

The chemical and microstructural features of the nanomaterials are presented in Figure 2a,b. From the spectra in Figure 2a, the characteristic peaks of B 1s and N 1s binding energies are identified for h-BN at 191.0 and 398.5 eV, respectively [21,22]. Curiously, one can also identify peaks associated with O 1s, and C 1s binding energies, at 533.2 and 285.2 eV, respectively. Even though the O 1s peak can appear due to oxidation processes, since this h-BN is not modified, it is likely that these two peaks can be ascribed to an incomplete removal of the solvent used in the exfoliation process, i.e., THF. As for the CNT spectrum, one can identify the survey characteristic peaks of C 1s at 285.0 eV, but also the O 1s, and N 1s at 532.8, and 398.4 eV, respectively [23]. In a similar manner to h-BN, since these CNT are not modified, the O 1s peak must be related to minimal surface oxidation and N 1s peaks can be related to the residual adsorbed solvent, considering that the CNT were debundled in a mixture of DMF/THF. After the “exfoliation” process of the hybrid sample, it was possible to identify all the previous peaks, which confirms the presence of both h-BN and CNT. Curiously, the intensity of the C 1s peak is much lower than the B 1s and N 1s peaks in the h-BN@CNT sample, probably because the measurement might have been performed in a h-BN rich region.

Regarding the microstructure of the studied nanomaterials, one can clearly observe in Figure 2b the characteristic h-BN diffraction peaks at 26.7°, 41.6°, 43.9°, 50.2°, and 55.2°. These peaks can be ascribed to the (200), (100), (101), (102), and (004) diffraction planes, respectively [1]. Differently, one cannot observe any well-defined peaks for the CNT, with only a very broad and low intensity halo at 25.9°, which would be related to the (002) plane of graphitic structures [24]. Regarding the hybrid h-BN@CNT, one can still observe all the characteristic peaks from h-BN. However, although the (100) and (004) peaks maintained their original intensities, the other three peaks seem to decrease in intensity, which might indicate some mixture effects of the CNT on the h-BN exfoliation.

### 3.3. Thermal Transport Performance

Thermal conductivity was evaluated for the base fluid, and the suspensions containing different wt.% of nano-additives. Improvements were observed as evaluations were performed at various temperatures. It is shown by many authors that an enhancement in thermal transport performance is achieved due to effects of nanostructures interacting with oil lubricant molecules [12]. The nanofluids thermal conductivity is governed by Brownian motions, as the collisions between nanostructures create a solid-to-solid conduction channel mode of heat transfer, which may increase due to the percolation process. Subsequently, thermal conductivity is increased by a convective heat transfer mode [25]. Additionally, the suspensions tend to be more stable when there are efficient interactions and compatibility between the nanoparticles with oil lubricants [26,27]. Figure 3a–c depicts the temperature-dependency behavior of the nanofluids’ thermal conductivity when filled by h-BN, CNT and the hybrid h-BN@CNT at various concentrations. The vegetable lubricant did not display significant temperature dependency (less than 1.5% at 85 °C). It must be mentioned that these types of lubricants are commonly used as coolant and dielectric materials in electrical devices, systems such as in power generators or high voltage power transmission systems (electrical transformers), which makes them suitable for relative high temperature oscillations, maintaining their properties and characteristics.

In general, the thermal conductivity of nanofluids was gradually increased for all investigated nano-additives. This trend occurred with increasing concentration and temperature, indicating improved physicochemical characteristics of thermal transport in fluids. Figure 3a shows the effect of h-BN on the thermal conductivity of the vegetable lubricant. As concentration and temperature were increased, an enhancement in thermal conductivity was also observed. For instance, at 50 °C, improvements of 9.2, 15.3 and 20.4% are obtained at 0.10, 0.25 and 0.50 wt.% of h-BN, respectively. As the testing temperature increased to 75 °C, it was possible to observe higher improvements of 13.7, 20.3 and 26.5%, respectively. Maximum enhancements of 18.2, 24.2 and 31.5% were observed when the evaluating temperature reached 85 °C. However, since above this temperature the measurements exhibited high variance due to fluid convection currents, which directly affect the measurements, this was the highest temperature investigated.

A similar improvement trend was observed for the nanofluids reinforced with CNT (Figure 3b). The CNT nanofluids showed higher enhancements than h-BN nanofluids. In this case, the observed improvement at 50 °C was 11.5, 18.9 and 21.8 at 0.10, 0.25 and 0.50 wt.%, respectively. As the evaluating temperature was raised to 75 °C, the CNT nanofluids showed 15.4, 23.1 and 27.8% increases in thermal conductivity compared to the base vegetable lubricant. The maximum evaluated temperature of 85 °C showed a maximum improvement of 19.9, 26.2 and 32.9%, respectively.

Thermal conductivity showed significant benefits when hybrid nanostructures were applied as fillers in the vegetable lubricant (Figure 3c). Here, the incorporation of h-BN@CNT nanostructures contributed to enhancements of 14.2, 19.9 and 23.7% at 0.10, 0.25 and 0.50 wt.% at room temperature, respectively. As the evaluating temperature was increased, the nanofluids exhibited higher performance. At 75 °C, the thermal conductivity of hybrid nanofluids achieved 18.8, 25.3 and 30.9%, respectively. At 85 °C, the maximum thermal conductivity performance of 21.4, 29.1 and 39.0% was observed at 0.10, 0.25 and 0.50 wt.%, respectively. It is important to mention that, as the evaluating temperature increased, more deviation was observed in the obtained data. This is due to the features of the lubricant and its interactions with the studied nanostructures. In previous research, Taha et al. [1] investigated the thermal conductivity of synthetic and natural esters, by reinforcing them with 2D nanostructures, including single h-BN, single molybdenum disulfide (MoS_2_) and a hybrid combination h-BN@MoS_2_. These nanosheets were incorporated into the lubricants. The nano-lubricants were prepared at various filler fractions by weight (0.10–0.25 wt.%), producing stable suspensions without the use of stabilizing agents or surfactants. Experimental results showed that the addition of 2D-nanostructures led to a 20–32% improvement in thermal conductivity for the h-BN@MoS_2_ hybrid.

### 3.4. Rheology Performance

The rheological characteristics are crucial to the final processing and application of a fluid. The introduction of nanomaterials in the base oil can result in suspensions with a rheological behavior different from the dispersant. Therefore, analyzing the rheological influence of h-BN, CNT and h-BN@CNT suspensions in the oil lubricant is necessary. Steady shear flow rheological experiments were conducted to obtain rheological information for the suspensions. Figure 4a–f presents the suspensions and base oils’ flow and viscosity curves. It is worth mentioning that the curves presented in Figure 4 that do not follow Newtonian behavior were treated by the Weissenberg–Rabinowitsch correction described in Equation (2) [28]:(2)$$\tau_{R}=\frac{2M}{\pi R^{3}}\left[\frac{3}{4}+\frac{1}{4}\frac{\partial ln(M)}{\partial ln({\dot{\gamma }}_{R})}\right]$$
where R is the plate radius, $\tau_{R}$ is the shear stress at R, M is the torque, and ${\dot{\gamma }}_{R}$ is the shear rate at R. This correction is necessary because, when using the geometry of parallel plates, the shear rate of the non-Newtonian fluid varies along the plate radius; that is, the viscosity would not be uniform.

Initially, it is essential to analyze the behavior of the base oil. The curves of the base lubricant show a Newtonian fluid, which has a constant viscosity as a function of shear rate. For a flow subject to simple shear, the Newtonian model is described by Equation (3) [29]:(3)$$\tau =\eta \dot{\gamma }$$
where τ is the shear stress, η is the viscosity and $\dot{\gamma }$ is the shear rate. When analyzing the curves of the h-BN suspensions in Figure 4a,b, it is possible to observe an equivalent behavior to that of the base oil. In other words, the Newtonian model also represents h-BN suspensions in the shear rate range analyzed. In this case, the increase in the nanosheets’ concentration did not cause significant effects on the rheology of the suspensions, which presented similar values of viscosity and shear stress, both among themselves and compared to the base oil. This result may be related to the planar geometry of the h-BN sheets, which allows them to be easily dragged along the flow direction. In addition, it can also indicate the lack of interaction of the nanosheets with the oil dispersant due to the different type of interactions between the base oil and the h-BN nanoparticles. h-BN has a partially ionic character, while the lubricating oil has a hydrophobic character, making the interaction between them difficult [30].

On the other hand, the curves of the CNT’ suspensions show that adding nanotubes promotes the modification of the rheological behavior. The graphs presented in Figure 4c,d indicate that the increase in the shear rate leads to a decrease in viscosity, denoting a pseudoplastic behavior. Such behavior is described by the Power Law model, or Ostwald-de Waele Equation, which for simple shear flow is described by Equation (4) [31]:(4)$$\tau =k{\dot{\gamma }}^{n}$$
where k is the consistency and n is the power-law index. Note that when n = 1, the equation becomes the Newtonian model, with k corresponding to the viscosity (η) of the fluid. Table 3 shows the parameters of the rheological models used for the base fluid and the suspensions.

Table 3 shows that increasing the concentration of CNT causes a drop in the values of n, showing the development of the pseudoplastic nature and consequent departure from the Newtonian model. In addition, an increase in k with an increase in CNT concentration supports an increase in viscosity, mainly at low shear rates.

The rheological behavior of the CNT suspensions reveals that the nanotubes affect the viscoelasticity of the system. Observed behavior can be attributed to the interaction with the base oil, as will be discussed later, and interaction among CNT due to the 1D geometry that leads to the formation of bundles. In fact, there are studies reported in the literature that indicate that the addition of CNT to oil lubricants can lead to the formation of such entanglements [14,32,33]. Raising the CNT concentration in the suspension can cause them to entangle and form larger aggregates. As a result, the entangled nanotubes in these aggregates can increase the suspension flow resistance, increasing viscosity. The rise in the shear rate can promote the breakage and/or disentanglement of the agglomerates and the orientation of the nanotubes. Consequently, the viscosity values fall with the increase in the shear rate.

On the other hand, it is pertinent to indicate the possibility that the CNT interact with the base oil. The CNT, formed by carbon atoms, have a strongly non-polar character. The lubricant oil, in turn, is composed of long hydrophobic chains. These are indications that these materials can develop relevant interactions, which may also contribute to the increase in viscosity with the increase in CNT concentration [2].

Figure 4e,f shows the curves of the h-BN@CNT suspensions. The results reveal that increasing the h-BN@CNT concentration in the system promotes an increase in viscosity, and there is a decrease in the apparent viscosity with the increase in the shear rate. Therefore, the suspensions have a stronger pseudoplastic behavior with content, which is also well represented by the Ostwald-de Waele model. As a result, there is a decrease in *n* and an increase in *k* with increasing concentration, as shown in Table 3.

Such results mainly show the impact of adding CNT to the system, since 50% of the nanostructures that make up the hybrid suspension are CNT. When compared, the apparent viscosity values of the hybrid suspensions at 0.25 wt.% and 0.50 wt.% with the CNT suspensions of 0.10 wt.% and 0.25 wt.%, respectively, are close. In contrast, hybrid nanofluids with a concentration of 0.10 wt.% have an apparent viscosity slightly lower than the base lubricant. This decrease in the viscosity of the hybrid suspension (Figure 4f) may be essentially related to the presence of h-BN. When we observe the viscosity and shear stress behaviors of the nanofluids produced only by individual h-BN nanoparticles (Figure 4a,b), they practically remain similar to the base fluid. This behavior was verified in some works in which the flow can be oriented in the presence of 2D nanomaterials [34]. Therefore, h-BN can assist in the debundling of CNT, which facilitates the flow. However, with an increase of CNT concentration, this process is no longer observed. This fact may be associated with the competition between the h-BN effect to maintain the Newtonian behavior and the CNT entanglement. The low content may have been responsible for this synergistic effect, in which the presence of h-BN significantly hindered the entanglement of the CNT and, therefore, the CNT could orient themselves more easily in the direction of flow, reducing the viscosity.

## 4. Discussion

Different types of nanofluids were prepared and characterized by individual components and 2D mixtures that act synergistically. Boron nitride and CNT contribute to increase thermal conductivity, and CNT also affect the base oil rheological properties. The thermal conduction mechanisms of a nanofluid filled with both CNT and h-BN are particularly intriguing due to the synergistic effects of these two nanomaterials (Figure 5a–c). Even though CNT are renowned for their exceptional one-dimensional thermal conductivity, and exhibit a hydrophobic character, which improves its interactions with the vegetable oil, CNT often tend to aggregate, limiting their dispersion and heat transfer capabilities in a fluid medium, as is also presented in Figure 5b. Here is where h-BN plays a crucial role. When h-BN nanosheets are introduced into the CNT nanofluid, h-BN not only adds to the overall heat conduction due to its high thermal conductivity in the basal plane, but also acts as a “stabilizing agent”, effectively debundling and bridging the CNT. This effect is proposed in Figure 5c. Thus, CNT are more easily separated and suspended in the fluid after the h-BN addition, preventing them from clumping together. This debundling effect can enhance the surface area available for thermal conduction, leading to improved heat transfer efficiency. As a result, the combination of CNT and h-BN in the nanofluid creates a dual mechanism for superior thermal conduction. CNT establish a conductive network for rapid heat transfer, while h-BN mitigates aggregation issues, allowing for better dispersion and heat transfer within the fluid. This combined effect makes the nanofluid an attractive option for advanced thermal management applications where efficient heat dissipation is critical.

This is an indication of the role of CNT in the general behavior of the hybrid system, as their component alone does not contribute to a similar increase in thermal conductivity. As a result, when the hybrid h-BN@CNT nanoparticles were evenly dispersed within the vegetable lubricants, they induced a larger impact on thermal conductivity, indicating high potential for multifunctional properties within advanced material applications. This type of synergistic behavior between different nanoparticles has already been observed in several studies [17,35], and this aspect should be highlighted. The key findings of this study include a remarkable enhancement in thermal conductivity, with increases of up to 39% observed in fluids containing 0.5 wt.% of hybrid nanomaterials. Regarding rheological properties, it was observed that both the base fluid and h-BN suspensions displayed Newtonian behavior, whereas the presence of CNT altered this behavior due to their hydrophobic nature. h-BN nanostructures were noted for their partial ionic character, attributed to lip–lip “bonds.” However, the most significant contribution to improved thermal conductivity was seen when combining these nanostructures, indicating a synergistic effect that outperformed individual counterparts. This novel approach to enhancing thermal and rheological properties in nanofluids represents a noteworthy advancement in the field.

## 5. Conclusions

Vegetable oil-based nanofluids, which are environmentally friendly, were developed by reinforcing them with h-BN, CNT and hybrid h-BN@CNT nanostructures at various filler fractions. Temperature dependent evaluations were performed to determine the thermal transport characteristics of the produced nanofluids. It was observed that thermal transport was influenced by the chemical and physical characteristics of the vegetable lubricant and its interactions with the nanostructures. Overall, the thermal conductivity performance of the nanofluids exhibited a temperature dependence, highlighting the significance of the interactions between the nanoparticles and the vegetable lubricant. The homogeneous dispersion by extensive ultrasonication and incorporation of the nanostructures within conventional lubricant achieved significant positive benefits on the effective thermal conductivity.

The thermal conductivity was improved by ~21% at 50 °C for single reinforcement of h-BN or CNT at 0.50 wt.%. Furthermore, for the hybrid h-BN@CNT nanofluids, an enhancement of 23% was observed. As the evaluation temperature was increased, the improvement in the thermal conductivity was also increased. At 75 °C, thermal conductivity for the single reinforced nanofluids was enhanced by ~25% at 0.50 wt.%, whereas for hybrid nanofluid at the same weight fraction, the thermal conductivity was increased to 27%. Finally, at 85 °C, nanofluids reinforced with single nanostructures showed an improvement in the range of 27–28%. Meanwhile, the hybrid nanofluid showed an improvement of about 39% at the highest nanofiller concentration studied. Therefore, the synergistic effects observed by the combination of different nanoparticles, leading to the increase in thermal conductivity and changes in the rheological behavior of nanofluids, can be further explored and optimized.

## Figures and Tables

**Figure 1 nanomaterials-13-02739-f001:**
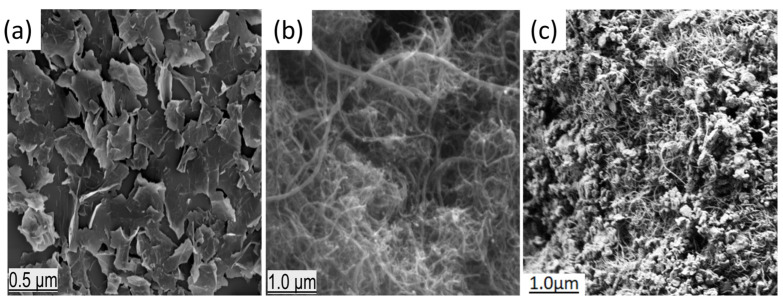
SEM images of h-BN (**a**) and CNT (**b**) and hybrid mixture h-BN@CNT (**c**) nanoparticles with different magnifications.

**Figure 2 nanomaterials-13-02739-f002:**
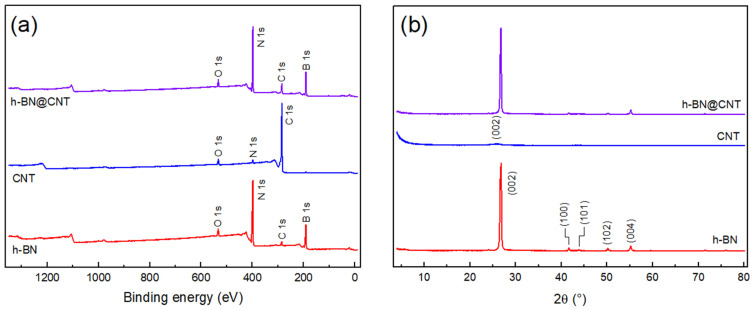
Chemical and microstructural analyses of h-BN, CNT, and h-BN@CNT: (**a**) XPS surveys; (**b**) XRD diffractograms.

**Figure 3 nanomaterials-13-02739-f003:**
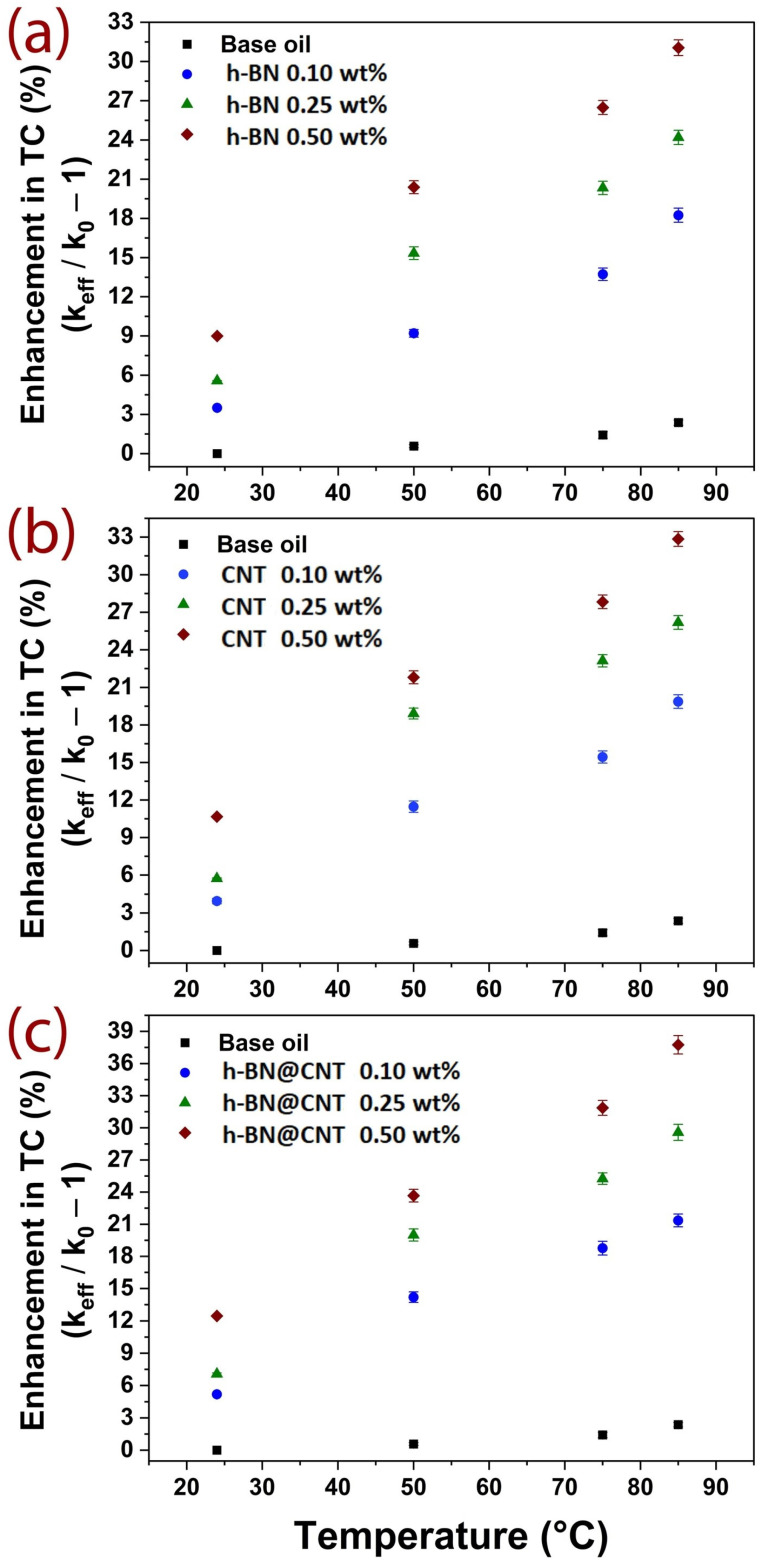
Thermal conductivity performance of vegetable nanofluids: (**a**) h-BN, (**b**) CNT, and (**c**) h-BN@CNT under temperature-dependence evaluation (percentage of filler amount is mentioned).

**Figure 4 nanomaterials-13-02739-f004:**
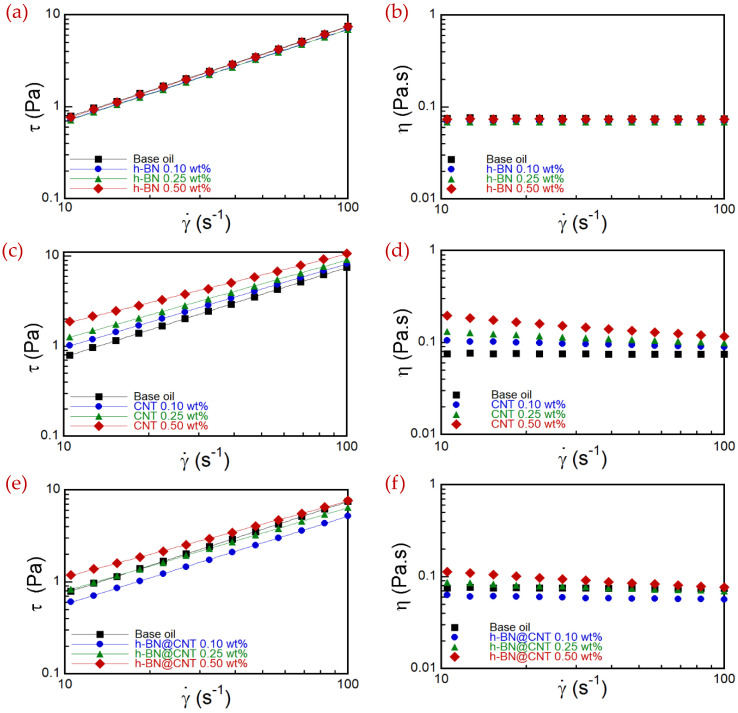
Flow curves along with curve fitting (left column) and viscosity curves (right columns) of the suspensions. All graphs show base oil curves as a reference. (**a**,**b**) present the effects of h-BN concentration; (**c**,**d**) show the influence of CNT concentration; (**e**,**f**) exhibit the effects of the hybrid concentration of h-BN@CNT on the flow and viscosity curves of the suspensions.

**Figure 5 nanomaterials-13-02739-f005:**
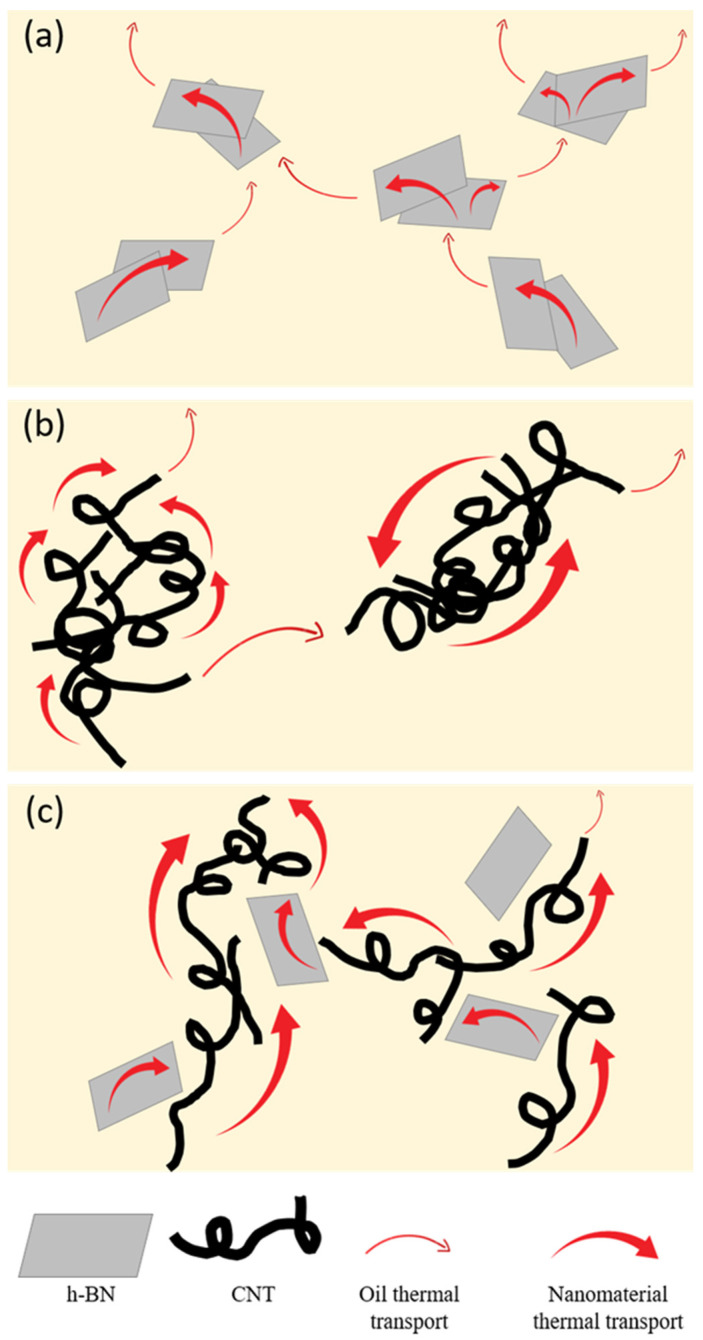
Illustrations of the proposed thermal conduction mechanisms within each nanofluid system: (**a**) h-BN, (**b**) CNT, and (**c**) h-BN@CNT.

**Table 1 nanomaterials-13-02739-t001:** Oil Properties [1].

General Properties	Standard		Units
Density at 20 °C	ISO 3675	<0.96	g/cm^3^
Viscosity at 40 °C	D445	<50	mm^2^/s
Dissipation Factor 25 °C	D924	<0.20	-
Acute Toxicity	OECD 202	Non-Toxic	-

**Table 2 nanomaterials-13-02739-t002:** Nanostructures characteristics at 298 K [1,19,20].

General Properties	h-BN	MWCNT	Units
Purity	98	>95	%
Density	2.29	1.7–2.1	g/cm^3^
Diameter	-	8–15	nm
Particle Size/Length	~1	10–50	µm
Thermal conductivity	300–500	3000–2000	W/m K

**Table 3 nanomaterials-13-02739-t003:** Parameters of the corresponding rheological models for each suspension. For suspensions with Newtonian behavior, k = η.

Suspension	Nanomaterial Concentration (wt.%)	K (Pa.s^n^)	n	Model
Base Oil	0	0.075	1	Newtonian
h-BN	0.10	0.070	1	Newtonian
	0.25	0.069	1	Newtonian
	0.50	0.074	1	Newtonian
CNT	0.10	0.113	0.93	Ostwald-de Waele
	0.25	0.160	0.87	Ostwald-de Waele
	0.50	0.299	0.77	Ostwald-de Waele
h-BN@CNT	0.10	0.063	0.96	Ostwald-de Waele
	0.25	0.099	0.91	Ostwald-de Waele
	0.50	0.169	0.82	Ostwald-de Waele

## Data Availability

Not applicable.

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
