# Peer review of "Thermal Transport and Rheological Properties of Hybrid Nanofluids Based on Vegetable Lubricants"

_nanomaterials, 2023, doi:10.3390/nano13202739_

Round 1

Reviewer 1 Report

-Clearly explain the line of science, what has been done in this area, what is new in this paper.

-This paper is more like a report

- provide a comprehensive literature review about hybrid thermal conductivity and viscosity and demonstrate and compare graphically

N/A

Author Response

Good day,

Here we attach a document with our response to Reviewer 1.

Regards,

Reviewer 2 Report

1. How the h-BN particles separate the CNTs bundles apart in Fig. 1, and the SEM magnification of nanoparticles is inconsistent.

2. The text size in the figure is too large, the text size and the drawing name should be consistent.

3. “Without using any surfactant agents, the nanostructures were dispersed in isopropanol (IPA) via ultrasonication to produce the h-BN, CNTs, and h-BN@CNTs samples.” Whether the nanofluids be stable without the use of surfactants? Is there any influence on the measurement of data during the experiment.

4. “Regarding the microstructure of the studied nanomaterials, one can clearly observe in Figure 1b the characteristic h-BN diffraction peaks at 26.7°, 41.6°, 43.9°, 50.2°, and 55.2°. Is it observed in Fig. 1?

5. What are the thermal conduction mechanisms in Fig. 5?The article does not explain Fig. 5.

Minor editing of English language required

Author Response

Good day,

Here we attach a document with our response to Reviewer 2.

Regards,

Round 2

Reviewer 1 Report

N/A

fine